# An Integrated Multi-Omic Network Analysis Identifies Seizure-Associated Dysregulated Pathways in the GAERS Model of Absence Epilepsy

**DOI:** 10.3390/ijms23116063

**Published:** 2022-05-28

**Authors:** Anna Harutyunyan, Debbie Chong, Rui Li, Anup D. Shah, Zahra Ali, Cheng Huang, Christopher K. Barlow, Piero Perucca, Terence J. O’Brien, Nigel C. Jones, Ralf B. Schittenhelm, Alison Anderson, Pablo M. Casillas-Espinosa

**Affiliations:** 1Department of Medicine, The Royal Melbourne Hospital, The University of Melbourne, Melbourne, VIC 3052, Australia; aharutyunyan@student.unimelb.edu.au (A.H.); te.obrien@alfred.org.au (T.J.O.); nigel.jones@monash.edu (N.C.J.); alison.anderson1@monash.edu (A.A.); 2Department of Neuroscience, Central Clinical School, Monash University, Melbourne, VIC 3004, Australia; debbie.chong@monash.edu (D.C.); rui.li@monash.edu (R.L.); zahra0016@gmail.com (Z.A.); piero.perucca@monash.edu (P.P.); 3Monash Biomedical Proteomics Facility and Monash Biomedicine Discovery Institute, Monash University, Clayton, VIC 3800, Australia; anup.shah@monash.edu (A.D.S.); cheng.huang@monash.edu (C.H.); chris.barlow@monash.edu (C.K.B.); ralf.schittenhelm@monash.edu (R.B.S.); 4Department of Neurology, Alfred Health, Melbourne, VIC 3004, Australia; 5Bladin-Berkovic Comprehensive Epilepsy Program, Department of Neurology, Austin Health, Heidelberg, VIC 3084, Australia; 6Epilepsy Research Centre, Department of Medicine, Austin Health, The University of Melbourne, Heidelberg, VIC 3084, Australia

**Keywords:** absence epilepsy, ALDH2, GAERS, GSTM1, lysine degradation, metabolomics, proteomics, WGCNA

## Abstract

Absence epilepsy syndromes are part of the genetic generalized epilepsies, the pathogenesis of which remains poorly understood, although a polygenic architecture is presumed. Current focus on single molecule or gene identification to elucidate epileptogenic drivers is unable to fully capture the complex dysfunctional interactions occurring at a genetic/proteomic/metabolomic level. Here, we employ a multi-omic, network-based approach to characterize the molecular signature associated with absence epilepsy-like phenotype seen in a well validated rat model of genetic generalized epilepsy with absence seizures. Electroencephalographic and behavioral data was collected from Genetic Absence Epilepsy Rats from Strasbourg (GAERS, *n* = 6) and non-epileptic controls (NEC, *n* = 6), followed by proteomic and metabolomic profiling of the cortical and thalamic tissue of rats from both groups. The general framework of weighted correlation network analysis (WGCNA) was used to identify groups of highly correlated proteins and metabolites, which were then functionally annotated through joint pathway enrichment analysis. In both brain regions a large protein-metabolite module was found to be highly associated with the GAERS strain, absence seizures and associated anxiety and depressive-like phenotype. Quantitative pathway analysis indicated enrichment in oxidative pathways and a downregulation of the lysine degradation pathway in both brain regions. GSTM1 and ALDH2 were identified as central regulatory hubs of the seizure-associated module in the somatosensory cortex and thalamus, respectively. These enzymes are involved in lysine degradation and play important roles in maintaining oxidative balance. We conclude that the dysregulated pathways identified in the seizure-associated module may be involved in the aetiology and maintenance of absence seizure activity. This dysregulated activity could potentially be modulated by targeting one or both central regulatory hubs.

## 1. Introduction

The epilepsies are a heterogenous group of chronic neurological conditions characterized by recurrent spontaneous seizures and often associated with psychiatric and cognitive comorbidities [1] They are estimated to affect over 50 million people worldwide and make up 0.5% of the global burden of disease [2,3].

Absence epilepsy syndromes, including childhood absence epilepsy and juvenile absence epilepsy, are grouped under the genetic generalized epilepsy (GGE) umbrella by the latest classification of the International League Against Epilepsy (ILAE) [4]. While some GGE cases have been linked to a single causal gene, the vast majority are presumed to have a polygenic architecture [5,6]. This assumption is supported by observations that most patients with GGE do not have a strong family history of epilepsy and that siblings of affected individuals have an 8% risk of developing epilepsy, which is lower that the risk expected for a recessive (25%) or dominant (50%) inherited trait [7,8].

The pathophysiological changes leading to absence epilepsy have been studied in various animal models [9,10]. One of the most utilized and validated is the Genetic Absence Epilepsy Rats from Strasbourg (GAERS) model, derived by selective inbreeding of Wistar rats that expressed spontaneous absence-type seizures accompanied by generalized spike and wave discharges on EEG recordings [11,12,13]. The GAERS model shares many behavioral characteristics and pharmacological response profile to anti-seizure medication to human absence epilepsy [11,14,15,16]. Absence epilepsy involves diffuse, bilateral cerebral regions and predominantly presents in children and adolescents [17] and as such, resective epilepsy surgery is unable to be conducted on these individuals. Therefore, human tissue collection from living individuals for research is challenging. To this end, animal models, such as the GAERS, are especially important to aid the investigation of novel targeted therapy development and have high potential to elucidate the molecular changes underlying disease development. Nevertheless, many studies aiming to characterize the molecular profile of these animals have generally been constrained to a few molecules, or single “omes” such as the genome [18] or proteome [19,20]. These single layer studies are limited to only providing details on the dysregulated genes/proteins without any indication about potential inter-layer interactions, which are especially pertinent to complex diseases such as epilepsy [21]. Identifying causative genetic markers is challenging due to polygenic architecture of the GGEs. Additionally, numerous changes can occur after initial gene expression, such as epigenetic modifications, that can influence downstream protein expression and the function of larger network of biological pathways [22]. Thus, the proteome and the metabolome can better reflect the observed phenotype, as they provide snapshots of all processes that are underway at any given time [23]. At present, there is a lack of studies investigating changes at these two levels in the context of absence epilepsy. We envisaged that integration of these layers would enable the discovery of coherent molecular signatures, and identify cellular mechanisms and changes, not present at an earlier stage of molecular expression, that are more relevant to disease development.

Biological systems respond to multiple inputs that vary simultaneously and interact with each other in multiple ways, forming complex molecular networks. As such, the use of systems biology and network theory methodology to model and characterize these networks can elucidate the emergent behavior of the system. Over the past decade, there has been growing support and evidence for changing the scientific paradigm of disease diagnosis and treatment from “single gene–single disease” to a more holistic systems approach, wherein groups/modules of genes (as well as gene products and metabolites) and the biological pathways they represent define the pathology. Network approaches have been used to identify these “disease modules”, study the pathophysiology mechanisms in a wide range of fields [24,25,26,27], and uncover molecular relationships between apparently distinct disease phenotypes [28]. Their utility in investigating epilepsy pathology has also been demonstrated [29,30]. Here, we aimed to: (a) gain insight into the biological pathways that drive the electrical and behavioral abnormalities seen in absence epilepsy, and (b) to identify the molecular signature associated with absence epilepsy in the GAERS model. To this end, we conducted behavioral analysis and electroencephalographic (EEG) profiling of the GAERS in relation to non-epileptic control (NEC) rats, followed by mass spectrometry-based proteomic and metabolomic analysis of the brain tissue from both groups. Given that absence epilepsy is a disorder of corticothalamic networks [31], we selected the somatosensory cortex (SCx) and thalamus, key regions in this network, for proteomic and metabolomic analyses. Utilizing basic statistical techniques for multisensor data merging and the framework of weighted correlation network analysis, we integrated the data from two molecular layers into multi-omic networks, which were then clustered into modules of highly correlated proteins and metabolites. We then assessed the correlation of these protein-metabolite modules to behavioral and seizure outcomes from the GAERS and NEC rats, with the aim of identifying modules showing strong correlation with the seizure and behavioral phenotype of GAERS.

## 2. Results

### 2.1. Behavioural Testing and Electroencephalography (EEG) Recordings Confirm Epileptic Phenotype in the GAERS Group

EEG analysis showed that twenty-four-week-old male GAERS experienced on average, around 200 seizures in the 24-h EEG period, each lasting for ~7 s on average (Figure 1a), while the age and sex matched NEC rats did not display any seizures (Figure 1a–c). Behavioral testing also confirmed that the GAERS exhibited increased anxious behavior, measured by the open field test (OFT). Compared to the NECs, the GAERS had decreased number of entries (*p* = 0.037) and time spent in the center of the open field (*p* = 0.008) (Figure 1d). Further, the GAERS also showed a decreased preference for sucrose compared to NECs (*p* = 0.026, Figure 1e), which is indicative of anhedonic-like behavior.

### 2.2. Proteomic Analysis Identifies Various Differentially Expressed Proteins in the GAERS Group 

We used untargeted proteomics to identify differentially expressed proteins in GAERS and NEC rats. In the SCx, 102 differentially expressed proteins were identified. From these, 55 proteins were found to be upregulated in GAERS compared to NEC, while 47 were downregulated (Figure 2a, Appendix A). In the thalamus, 123 proteins were identified as differentially expressed (Figure 2b, Appendix A), with 74 upregulated and 49 downregulated in GAERS compared to the NEC. The top 10 differentially expressed proteins in the SCx and thalamus are listed in Table 1 and Table 2, respectively. Amongst the differentially expressed proteins, 53 were common between the two cerebral regions (Figure 2c). Pathway Analysis with Down-weighting of Overlapping Genes (PADOG) indicated upregulation of pathways involved in oxidation and metabolism/clearance of neurotransmitters and their precursors in both brain regions (Figure 2d,e).

### 2.3. Metabolomic Analysis Identifies Differentially Abundant Metabolites and Significantly Enriched Metabolic Pathways in the GAERS Group

Similar to proteomics, untargeted metabolomics data were acquired using high-resolution mass spectrometry (MS) from the same brain regions of GAERS and NEC. After initial filtering and normalization, 897 metabolites were included in the statistical analysis (criteria: Fold-change > 1.5; FDR < 0.05, Appendix A). In the SCx, 57 metabolites were found to have significantly different abundance between strains, amongst which 27 showed a decrease and 30 showed an increase in GAERS (Figure 3a). In the thalamus, 45 metabolites had significantly different abundance, with 23 decreased and 22 increased in GAERS compared to NEC (Figure 3b). Amongst the differentially abundant metabolites, 29 were common between the two cerebral regions (Figure 3c). Metabolite set enrichment analysis (MSEA) showed significant enrichment in several amino acid metabolic pathways, galactose metabolism, glycolysis, and lysine degradation in both brain regions (Figure 3d,e).

### 2.4. Modules with Varying Correlations to GAERS and Seizure Phenotype Identified in the Multi-Omic Networks from Somatosensory Cortex and Thalamus

To represent the proteome and metabolome of GAERS and NEC in the context of a network, we leveraged the general framework of Weighted Gene Coexpression Network Analysis (WGCNA) to integrate the proteomic and metabolomic datasets into two multi-omic correlation networks, each representing the respective brain region. In these networks, each node corresponds to a single molecule (metabolite/protein) and the edges between nodes represent the correlation between the relative abundance of the given molecule across all samples. A hierarchical clustering algorithm identified 22 distinct protein-metabolite modules of various size, density and connectivity (Figure 4a,c, Appendix A) in both SCx and thalamic networks. Each module of the given network was uniquely annotated by color. We then assessed the correlation strength of each module to the strain (GAERS vs. NEC), various seizure parameters and behavioral outcomes from rats in both groups. In the SCx network, the Blue module showed the strongest and most significant (PCC = 1, *p* = 2 × 10^−10^) correlation with the GAERS strain. It also showed strong correlation to longer average seizure duration (PCC = 0.98, *p* = 1 × 10^−7^) and higher seizure frequency (PCC = 0.81, *p* = 0.003) (Figure 4b). This module is functionally enriched for multiple pathways, including ‘aminoacyl-tRNA biosynthesis’, ‘ABC transporters’, and ‘protein digestion and absorption’ (Table 3). Glutathione S-transferase mu 1 (Gstm1) was identified as the central regulatory hub of this module. Gstm1 is one of the top upregulated proteins in SCx (FC = 4.6, FDR = 1.8 × 10^−8^) (Table 1). It is noteworthy that with the exception of three proteins, all of the differentially expressed proteins identified in the SCx, regardless of expression levels, belong to the Blue module. This suggests that the concerted action of these proteins and their coregulation is drastically altered in the brain of GAERS compared to NECs.

WGCNA analysis of the thalamic datasets revealed 22 modules (Figure 3c,d, Appendix A). Again, there was one module that was highly correlated with the GAERS (PCC = 1, *p* = 1 × 10^−12^). It was strongly correlated with longer average duration (PCC = 0.98, *p* = 2 × 10^−8^) and higher frequency of seizures (PCC = 0.83, *p* = 9 × 10^−4^). This module, co-incidentally also labelled Blue, was enriched for numerous pathways implicating synaptic signaling and plasticity, and the metabolism of various neurotransmitters and amino acids (Appendix A). Interestingly, several significantly enriched thalamic pathways including: ‘lysine degradation’, ‘ABC transporters’, and ‘aminoacyl-tRNA biosynthesis’, were also significantly enriched in the SCx (Table 3). Aldehyde dehydrogenase 2 family member (Aldh2) was found to be the central hub of this module. Aldh2 is one of the top upregulated proteins in the thalamus of GAERS (FC = 5.4, FDR = 7.34 × 10^−13^) and is involved in the metabolism of aldehydes produced by lipid peroxidation such as ethanol and 4-hydroxy-2-noneal (4-HNE) [32].

### 2.5. Seizure-Associated Modules Show Significant Overlap in SCx and Thalamus

To disentangle the region-specific and global signatures associated with the GAERS, we conducted an overlap analysis between all 22 modules from both cerebral regions to reveal common proteins and metabolites constituting each SCx and thalamic module. Notably, the most significant (*p* = 3.9 × 10^−46^) overlap with 189 proteins/metabolites was between the GAERS-associated Blue modules from the respective SCx and thalamic networks (Figure 5a). Joint pathway analysis of the overlapping proteins/metabolites revealed enrichment in several pathways, with “lysine degradation” and “Aminoacyl-tRNA biosynthesis” as the most significantly enriched (FDR = 0.0073, Figure 5b). Since the Blue modules in both networks show strong correlation with seizures, anxious and anhedonic-like behavior—all of which are characteristic of absence epilepsy, we conclude that these modules are the most significant determinants of the molecular signature associated with the absence seizure phenotype the GAERS exhibit. Therefore, our subsequent analyses focus on these two seizure-associated modules.

### 2.6. Quantitative Enrichment Analysis of the Seizure-Associated Modules Identifies Various Differentially Regulated Pathways

To identify specific seizure-associated dysregulated pathways, we carried out quantitative enrichment analysis on the proteins comprising the seizure-associated Blue modules in both the SCx and thalamus. Using the PADOG (Pathway Analysis with Down-weighting of Overlapping Genes) module of Reactome database [33,34] we identified over 700 differentially regulated pathways in both cerebral regions. The top upregulated pathways in the GAERS were involved in synthesis, transport and clearance of neurotransmitters, synaptic signaling, and oxidative processes (Figure 6a,b). Conversely, the pathways that were downregulated in the GAERS compared to the NEC were involved in lysine catabolism, GTPase cycle, breakdown of galactose and glycogen, necrosis regulation and the innate immune system (Figure 6a,b).

## 3. Discussion

In this study, we demonstrate the utility of a network-based, integrative multi-omics approach to interrogate the molecular signature associated with absence epilepsy. While traditional linear analysis of single-omics data provided large lists of differentially abundant proteins and metabolites, when the two omics layers were integrated into a network along with behavioral and EEG data relevant to the disease phenotype, a specific protein-metabolite module that defines the molecular signature of absence seizures was identified in each brain region. The strong correlation of these two Blue modules with the GAERS strain and more frequent seizures in both the SCx and thalamic correlation networks provide further support for its association with epilepsy and identify it as a “seizure-associated disease module”. Interestingly, almost all of the differentially expressed proteins and a large number of differentially abundant metabolites identified were represented in the seizure-associated Blue module in both regions. Overlap analysis of SCx and thalamic multi-omic networks revealed that the two seizure-associated modules have a large number of proteins and metabolites in common, and share numerous enriched pathways, indicating potential global mechanisms affected across both brain regions in epileptic animals. By conducting a joint pathway enrichment analysis which uses both proteins and metabolites in a single query, we generated a smaller list of enriched pathways which were further investigated. Additionally, we conducted quantitative pathway analysis (PADOG) through Reactome resource, which indicated the direction (upregulated vs. downregulated) of changes in the significant pathways. Reactome and PADOG were preferred over other pathway analysis types and resources due to several advantages. Firstly, Reactome is a manually curated database and thus, the included pathways have all been experimentally verified [34]. Secondly, the PADOG method addresses the issue of exaggerated significance assigned to genes/proteins which appear in a large number of pathways by assigning more weight to those that are gene set-specific. Thus, if the gene sets that are highly specific to the seizure-associated pathways are differentially regulated, it is more likely that these pathways are truly relevant to absence epilepsy pathology.

The majority of differentially regulated pathways from the seizure-associated modules indicate an increase in synaptic transmission and metabolism of neurotransmitters. Given the increased synchrony in neuronal firing in the context of absence epilepsy, these results are expected. However, the most notable characteristic feature of the molecular signature associated with GAERS elucidated by our analyses is the dysregulation of the lysine degradation pathway. Lysine degradation is the top commonly enriched pathway in the SCx and thalamus of the GAERS (Figure 5a). Mainly localized to mitochondria, lysine degradation provides substrates such as glutamate and acetyl-CoA for downstream metabolic cascades [35]. Currently, there are no links between absence epilepsy and lysine degradation dysregulation, however, perturbations in this pathway have been linked to pyridoxine-dependent epilepsy (PDE) [35,36]. Specifically, genetic mutations affecting ALDH7A1 activity (which is involved in cerebral lysine catabolism) have been identified as the genetic cause of PDE [36]. ALDH7A1 is part of the aldehyde dehydrogenase superfamily that includes ALDH2—the central regulatory hub of the thalamic seizure-associated module. Several of the substrates generated through lysine degradation are necessary for synthesis of Glutathione (GSH)—a tripeptide consisting of glutamate, cysteine and glycine (Glu-Cys-Gly). Glutathione is an important anti-oxidant in the mammalian brain that protects cells from damaging effects of free radicals by conjugating with various reactive oxygen species and electrophiles [37]. The conjugation of glutathione is catalyzed by Glutathione-S transferases (GSTs)—a family of phase II detoxification enzymes which includes GSTM1, the regulatory hub of the seizure associated module in the SCx [38,39]. According to our metabolomic data, the concentration of GSH (Glu-Cys-Gly) is significantly decreased (Log2FC = −2.3, FDR = 0.038) in the thalamus of GAERS, and shows a trend towards decrease in SCx (Log2FC = −0.13, FDR = 0.2). The reduced concentration of GSH in both the SCx and thalamus hint towards a potential perturbation of oxidative stress balance in absence epilepsy. We postulate that in response to dysregulated lysine catabolism and increased oxidative burden, a compensatory “rewiring” of the downstream signaling pathways occurs in the GAERS brain, which is modulated by ALDH2 and GSTM1—the regulatory hubs of the seizure associated modules. This is further supported by our findings that ALDH2 and GSTM1 were among the top differentially expressed proteins in the GAERS strain. We speculate that on one hand, due to downregulated lysine catabolism and accumulation of L-lysine, there is a compensatory increase in aldehyde dehydrogenase activity mediated by ALDH2. On the other hand, due to increased synchronization of neuronal firing and shortage of GSH, there is higher oxidative burden and accumulation of ROS in the GAERS brain, which leads to compensatory increase in glutathione-S-transferase (GST) activity, mediated by GSTM1 and other GSTs. While no association between GSTM1 and absence epilepsy has previously been identified, individuals with drug-resistant epilepsy and a defect in GSTM1 enzymatic activity have increased levels of lipid peroxidation markers, compared to non-epileptic controls and epileptic individuals with normal GSTM1 activity [40]. Oxidative stress is widely recognized as a contributor of epileptogenesis [41,42], and while we have recognized it as a defining feature of the molecular signature of absence epilepsy, oxidative stress could be both the cause and the consequence of pathophysiology development.

It has been shown previously that modularity is a conserved property of biological systems and the cellular functions are carried out by highly connected modules of genes, proteins and metabolites [43,44]. These functional modules tend to be extremely heterogeneous, wherein the majority of the nodes have relatively few connections with other nodes, while a few “hub” nodes are highly connected and therefore are considered important regulators of the given module [45]. GSTM1 and ALDH2 were identified as the regulatory hubs of the seizure-associated module in the SCx and thalamus of GAERS, and therefore have the potential of influencing the larger molecular network they regulate, making them potential biomarkers of absence epilepsy and promising candidates for pharmacological manipulation.

The approach we applied allowed us to evaluate modules informed by both proteomic and metabolomic data. Regrettably, we lack transcriptomic data for further omics integration analyses. It can be noted, however, that little correlation has been found between the proteome and the transcriptome due to various downstream modifications including post-translational modifications or alternative splicing. It is also important to note that most of the biological pathway databases used in this study were curated using information gleaned from human studies. As such, their translatability to a rat model of epilepsy is assumed, but not verified. However, due to the lack of available databases utilizing information obtained from rodent studies, these databases represent the best available resource, as they are continuously updated with the most relevant and accurate information available. While the results obtained from this study were promising, to the best of our knowledge, many of these pathways and molecules have not been described in the context of absence epilepsy. Therefore, further experimental validation is necessary to establish these pathways and associated proteins and metabolites as being implicated in absence epilepsy. Overall, our study identifies novel pathways and regulatory hubs with strong potential as candidate biomarkers and treatment targets for drug repurposing and development.

## 4. Materials and Methods

### 4.1. EEG Electrode Implantation Surgery

Twenty-four-week-old male GAERS (*n* = 6) and NEC (*n* = 6) rats underwent EEG electrode implantation surgery under aseptic technique as previously described [18,46]. Briefly, animals were anesthetized with isoflurane (Ceva isoflurane, Piramal Enterprises Limited, India), the fur was shaved from the skull and a single midline incision was made on the scalp [47]. Four burr holes were drilled through the skull without penetrating the dura, one on each side of the frontoparietal region (AP: ±1.7; ML: −2.5), and one on each side of the temporal region, (AP: ±5.6; ML: left 2.5) anterior to lambda. Epidural stainless-steel screw recording electrodes (EM12/20/SPC, Plastics One Inc., Roanoke, VA, USA) were screwed into each hole. Ground and reference epidural stainless-steel screw electrodes were implanted on each side of the parietal bone above the cerebellum. The recording electrodes were fixed in position using self-curing dental cement (VX-SC1000GVD5/VX-SC1000GMLLQ, Vertex Pharmaceuticals, St. Leonards, Australia). The incision was sutured, and buprenorphine was administered intraperitoneally (0.05 mg/kg, Indivior Australia).

### 4.2. EEG Acquisition and Analysis

Animals were connected to the EEG acquisition system 10 days after electrode implantation surgery using cables (M12C−363, Plastics One Inc., VA, USA) that allowed free movement around the cage [48]. EEG recordings were acquired continuously for 24 h using Profusion 3 software (Compumedics, Melbourne, Australia), unfiltered and digitized at 512 Hz. An automatic detection algorithm developed by our group was used to detect and quantify SWDs [49]. The detection algorithm decodes the frequency spectrum power of the EEG data, one of the most important characteristics of SWD patterns. The SWDs detection module, composed of graph neural network and recurrent neural network, aggregates information across both the brain connectivity network and EEG temporal sequence. After automatic detection was performed, the total number and duration of SWDs along with average SWD duration were computed. An EEG recording was defined as a seizure if the SWD had an amplitude 3-times the baseline with a frequency of 7–12 Hz and a duration of more than 0.5 s [18,46].

### 4.3. Behavioural Tests

To evaluate the behavioral comorbidities reported in GAERS, we used the widely validated open field test (OFT) and sucrose preference test (SPT) [48,50]. All tests were performed in a light-controlled (~110 lux), closed, quiet and clean room between 9 am and 5 pm. Animals had at least one hour to acclimatize to the room prior to testing. Testing was performed in a blinded manner to strain. The OFT is a 100 cm diameter circular arena, with an inner circle arena of 66 cm in diameter. For each test, the rat is placed gently into the center of the field and its behavioral activity filmed from above for 10 min. The distance travelled, and the entries and time spent in the inner circle were objectively assessed from the video feed using Ethovision software (Ethovision 3.0.15, Noldus, Wageningen, The Netherlands) [48,50,51]. The SPT was performed 48 h after OFT completion. Animals remained in their home cage throughout the testing period. One hour before testing, animals were given up to 0.5 mL of 2% sucrose to familiarize them to the taste. Animals were then presented with two bottles, one filled with tap water and the other with 2% sucrose solution for 24 h [48]. Bottle position was randomized to avoid position preference [48,50,52]. Total fluid intake and percentage preference for sucrose were recorded.

### 4.4. Statistical Analysis

Statistical analysis of behavioral data was conducted using GraphPad Prism 9 (GraphPad Software Inc., San Diego, CA, USA). Normal distribution was assessed using the Shapiro–Wilk test. Unpaired T-tests were used to compare the group means of the normally distributed data (% sucrose preference, Figure 1e). The Mann-Whitney test was used to compare the group means of non-normally distributed data (number of seizures (Figure 1b), seizure duration (Figure 1c), and time spent in the center of open field (Figure 1d)). Statistical significance in all cases was set to *p* ≤ 0.05.

### 4.5. Tissue Preparation

Animals were anesthetized using 5% isoflurane and subsequently euthanized via 150 mg/kg pentobarbitone sodium intraperitoneal injection (Lethabarb, Virbac, Milperra, Australia). The thalamus and SCx were rapidly harvested, snap frozen in liquid nitrogen and stored at −80 °C. Approximately 30 mg of tissue was cryogenically pulverized using a 12-well biopulverizer (BioSpec Products, Bartlesville, OK, USA Part number 59012MS) according to manufacturer’s instructions. The biopulverizer and pestles were cooled in liquid nitrogen. Frozen samples were added to a well and pulverized by sharply striking the pestle four to five times with a mallet. Powdered tissue was transferred into a cold Eppendorf tube. Resulting tissue was split into two portions for proteomic and metabolomic analysis.

### 4.6. Proteomic Analysis Using LC-MS/MS

30 mg of tissue from each region was processed. Powdered samples were lysed in 4% SDS, 100 mM Tris (pH 8.1, 95 °C, 10 min) and sonicated. Lysate was cleared by centrifugation (16,000× *g*, 10 min) and protein concentration was determined using Pierce™ BCA Protein Assay Kit (Thermo Scientific, Waltham, MA, USA). Equal amount of protein was denatured and alkylated using Tris(2-carboxyethyl)-phosphine-hydrochloride and 2-Chloroacetamide (final concentration of 10 mM and 40 mM, respectively), and incubated (95 °C, 5 min). Proteins were precipitated using chloroform/methanol followed by sequencing grade trypsin digestion (37 °C, overnight, enzyme to protein ratio of 1:100). Digestion was stopped by adding formic acid (concentration of 1%). Peptides were cleaned with BondElut Omix Tips (Agilent) and concentrated in a vacuum concentrator prior to MS analysis.

Using a Dionex UltiMate 3000 RSLCnano system equipped with a Dionex UltiMate 3000 RS autosampler, samples were loaded via an Acclaim PepMap 100 trap column (100 µm × 2 cm, nanoViper, C18, 5 µm, 100å; Thermo Scientific, Waltham, MA, USA) onto an Acclaim PepMap RSLC analytical column (75 µm × 50 cm, nanoViper, C18, 2 µm, 100å; Thermo Scientific, Waltham, MA, USA). Peptides were separated by increasing concentrations of 80% ACN/0.1% FA at a flow of 250 nL/min (158 min) and analyzed with QExactive HF mass spectrometer (Thermo Scientific, Waltham, MA, USA) operated in data-independent acquisition (DIA) mode. Sixty sequential DIA windows (isolation width: 10 *m*/*z*) were acquired (375–975 *m*/*z*) (resolution: 15,000; AGC target: 2 × 10^5^; maximum IT: 9 ms; HCD Collision energy: 27%) following a full ms1 scan (resolution: 60.000; AGC target: 3 × 10^6^; maximum IT: 54 ms; scan range: 375–1575 *m*/*z*). Acquired DIA data were evaluated in Spectronaut 13 Laika (Biognosys) using in-house spectral library derived from the same brain samples. Multi-dimensional scaling was undertaken to identify outliers. Differential expression analysis was conducted using R Studio (version 1.2.5033) and R’s (version 3.6.3) limma package on log-transformed intensity values [53]. The raw mass spectrometry data has been deposted to the ProteomeXchange Consortium via the PRIDE partner repository [54] with the dataset identifier PXD033987. 

### 4.7. LC-MS Untargeted Metabolomic Analysis

Remaining pulverized frozen tissue was weighed by transferring to a fresh eppendorf tube and 20 µL of extraction solvent (2:6:1 CHCl_3_:MeOH:H_2_O *v*/*v*/*v*, internal standards: 2 µM CHAPS, CAPS, PIPES and TRIS) (0 °C) per mg of tissue was immediately added. The mixture was briefly vortexed before sonication in an ice-water bath (10 min) followed by centrifugation (20,000× *g*, 4 °C, 10 min). Supernatant was transferred to a MS vial for analysis. A Dionex RSLC3000 UHPLC coupled to a Q-Exactive Orbitrap MS (Thermo) was used. Samples were analyzed by hydrophilic interaction liquid chromatography (HILIC) following a previously published method [55]. The chromatography utilized a ZIC-p(HILIC) column 5 µm 150 × 4.6 mm with a 20 × 2.1 mm ZIC-pHILIC guard column (both Merck Millipore, Bayswater, Australia) (25 °C). A gradient elution of 20 mM ammonium carbonate (A) and acetonitrile (B) (linear gradient time-%B: 0 min-80%, 15 min-50%, 18 min-5%, 21 min-5%, 24 min-80%, 32 min-80%) was utilized. Flow rate was maintained at 300 μL/min. Samples were kept in the autosampler (6 °C) and 10 μL was injected for analysis. MS was performed at 35,000 resolution, operating in rapid switching positive (4 kV) and negative (−3.5 kV) mode electrospray ionization (capillary temperature 300 °C; sheath gas flow rate 50; auxiliary gas flow rate 20; sweep gas 2; probe temp 120 °C). Samples were randomized and processed in a single batch with intermittent analysis of pooled quality-control samples to ensure reproducibility and minimize variation. For accurate metabolite identification, a standard library of ~300 metabolites were analyzed before sample testing and accurate retention time for each standard was recorded. This standard library also forms the basis of a retention time prediction model used to provide putative identification of metabolites not contained within the standard library [56]. Acquired LC-MS/MS data was processed in an untargeted fashion using open source software IDEOM, which initially used *ProteoWizard* to convert raw LC-MS files to *mzXML* format and *XCMS* to pick peaks to convert to *peakML* files [57]. *Mzmatch.R* was subsequently used for sample alignment and filtering [58]. IDEOM was utilized for further data pre-processing, organization and quality evaluation [57]. Raw peak intensity values of metabolites which passed the RT error check were included in the final data matrix for statistical analysis. Peak intensity values were log-transformed, quantile normalized and unit-variance scaled to achieve normal distribution. Principal component analysis was conducted to identify and remove outliers prior to further statistical analysis. To identify differences in metabolite abundance between strains, the fold changes of each metabolite were calculated and compared via unpaired *t*-test between the groups. For both proteomics and metabolomics, *p*-values associated with the t-tests were corrected for multiple comparisons using the Benjamini-Hochberg method, and significance threshold was set to FDR < 0.05 [59].

### 4.8. Multi-Omic Data Integration and Weighted Gene Co-Expression Network Analysis (WGCNA)

To integrate proteomic and metabolomic data for network analysis, we unit-variance scaled and concatenated the normalized data into a single matrix for each cerebral region. Unit-variance scaling uses standard deviation as the scaling factor, thus, the resultant integrated data can be analyzed based on correlations [60]. Correlation-based multi-omic networks were then constructed by employing the framework of WGCNA. An adjacency matrix was constructed reflecting the pairwise Pearson correlations between all detected proteins and metabolites across all samples in each dataset. Correlation networks for each brain region were built based on respective adjacency matrices. In these networks, each node corresponds to a single molecule and the edges between nodes represent the correlation between the relative abundance of the given metabolite/protein across all samples. An average linkage hierarchical clustering algorithm was employed to identify metabolite/protein modules (arbitrarily labelled by color). Central regulatory hubs were determined for each module by identifying the node with highest degree centrality (largest number of connections to other nodes) and most significant correlation to the first principal component associated with the module. For each network, the correlation of modules to the GAERS strain, seizure phenotype and cognitive performance was assessed through Pearson correlation. To assess overlaps in module composition of the networks representing each cerebral region, a cross-tabulation based approach was employed to generate contingency tables reporting the number of overlapping proteins/metabolites [61]. Overlap significance (whether the number of overlapping proteins/metabolites is larger than expected by chance) was assessed through Fisher’s exact test.

### 4.9. Enrichment Analysis

Enrichment analysis was carried out using various publicly available web tools and packages implemented in the R/RStudio environment. For single-omics enrichment analysis, MetaboAnalyst 5.0’s quantitative enrichment analysis module was used [62] for metabolomic data and the PADOG (Pathway Analysis with Down-weighting of Overlapping Genes) module of Reactome [33,34] was used for proteomic data. To functionally annotate the protein-metabolite modules identified by WGCNA we employed the joint pathway analysis module of MetaboAnalyst 5.0 [62]. This analysis uses both proteins and metabolites in a single query and is based on weighted data integration to address the issue of genes/proteins overwhelming the integrated pathway analysis results due to significantly different sizes of genomic/transcriptomic and metabolomic pathway databases.

## Figures and Tables

**Figure 1 ijms-23-06063-f001:**
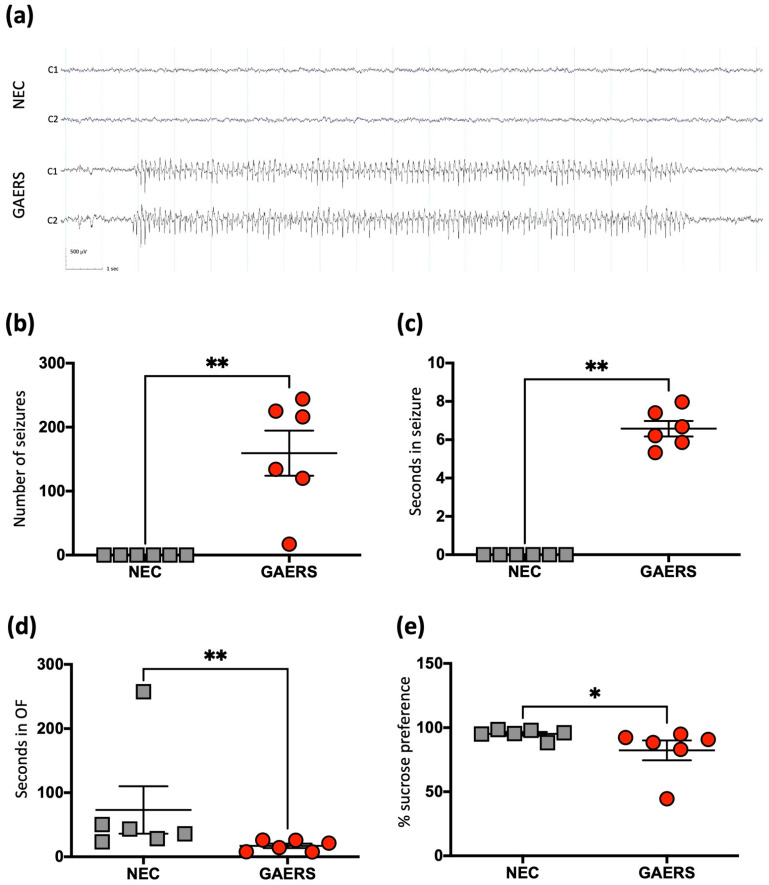
The EEG profile, seizure and behavioral outcomes observed in GAERS and NEC groups. (**a**) A recorded example of the characteristic EEG trace of the NEC and GAERS rats. (**b**–**e**) A two-tailed *t*-test was used for all comparisons, data shown as mean with standard error of the mean (SEM), significance indicated with asterisks: (*) at *p* < 0.05 and (**) at *p* < 0.01; (**b**) The number of seizures observed for individual rats. (**c**) The average time spent in seizures for individual rats. (**d**) The amount of time spent in the center of the open field. (**e**) The percentage of sucrose preference.

**Figure 2 ijms-23-06063-f002:**
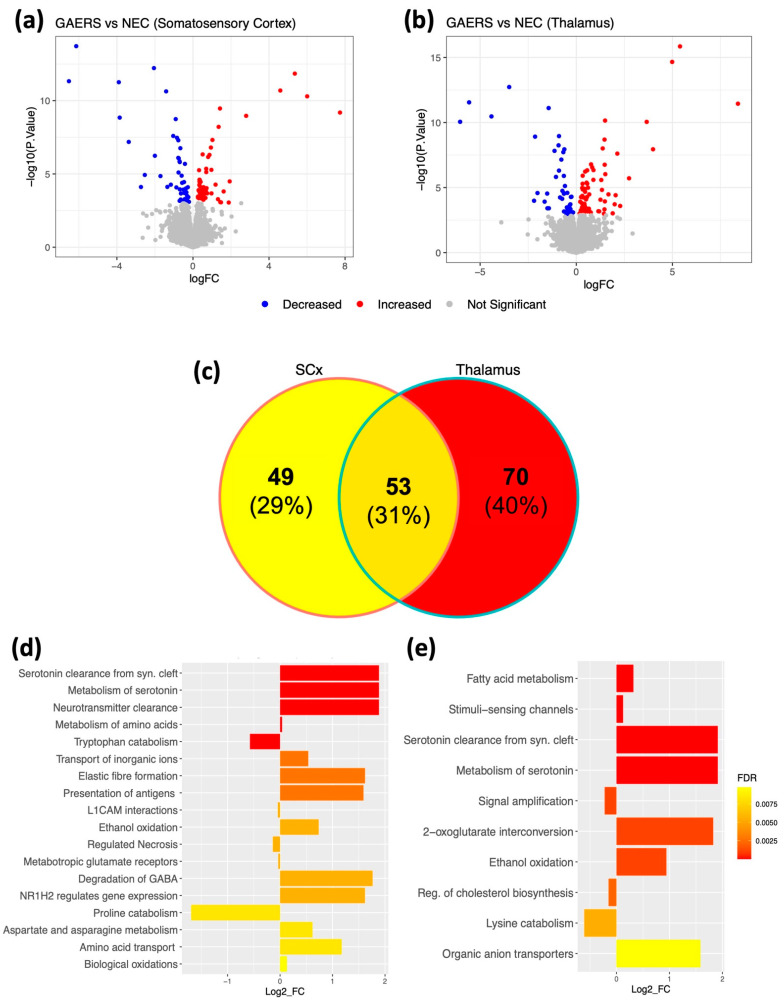
Proteomic profiling of the somatosensory cortex (SCx) and thalamus of GAERS compared to NEC. Top: Volcano plots showing the changes in the abundance of all quantified proteins (log_2_FC, *x*-axis) and their statistical significance (−log_10_ (*p*-value), *y*-axis) in GEARS relative to NEC in the (**a**) SCx and (**b**) thalamus. The upregulated proteins in GAERS compared to NEC are shown in red and downregulated proteins are in blue. (**c**) Venn diagram showing the differentially expressed proteins that are region-specific or common between SCx and thalamus of GAERS. Bottom: Differentially regulated pathways identified in the (**d**) SCx and (**e**) thalamus of GAERS, based on the PADOG analysis. The log_2_FC (*x*-axis) indicates the direction of regulation (log_2_FC > 0 upregulated, log_2_FC < 0 downregulated) of the whole pathway in GAERS relative to NEC.

**Figure 3 ijms-23-06063-f003:**
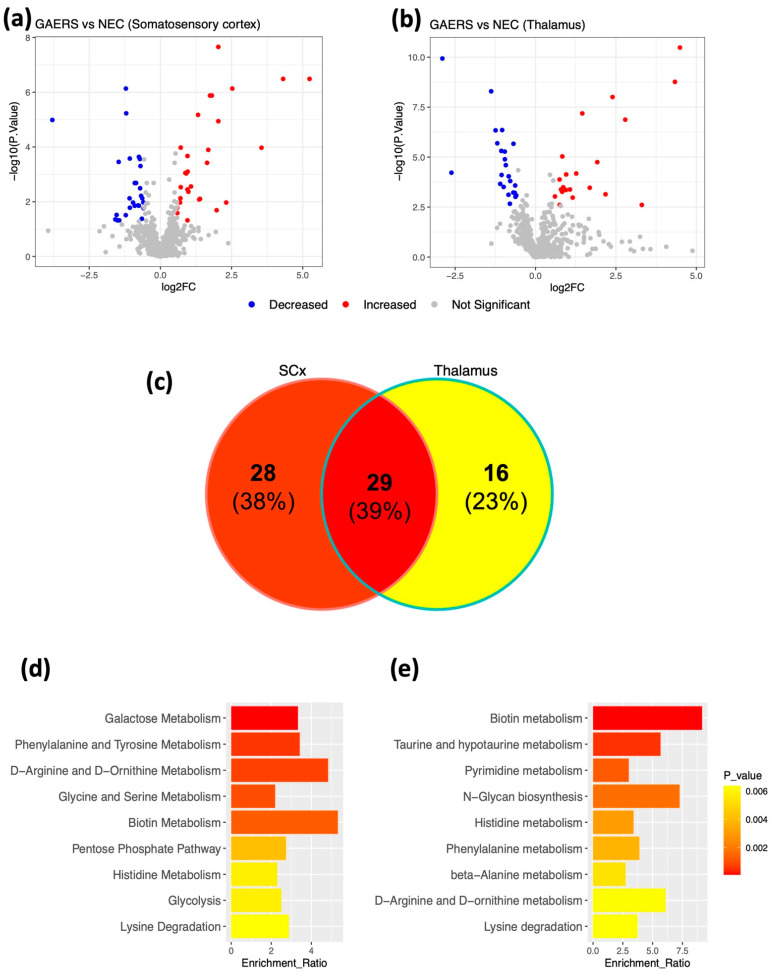
Metabolomic profiling of the SCx and thalamus of GAERS compared to NEC. Top: Volcano plot of the −Log_10_ (*p* value) vs. Log_2_FC of all identified metabolites in GAERS relative to NEC in the (**a**) somatosensory cortex and (**b**) thalamus with the metabolites showing increased abundance in GAERS compared to NEC in red and those with decreased abundance in GAERS compared to NEC in blue. (**c**) a Venn diagram showing the differentially abundant metabolites that are region-specific or common between SCx and thalamus of GAERS. Bottom: The enriched pathways identified in the (**d**) SCx and (**e**) thalamus of GAERS, based on the metabolite-set enrichment analysis (MSEA).

**Figure 4 ijms-23-06063-f004:**
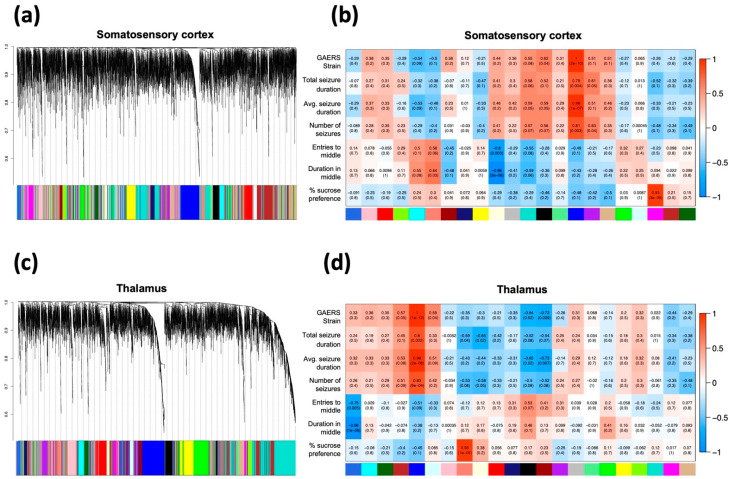
**WGCNA of integrated multi-omic data from somatosensory cortex and thalamus.** Left: Hierarchical cluster dendrogram of (**a**) SCx and (**c**) thalamic correlation networks. Each vertical line in the dendrogram represents a single protein/metabolite, with the arbitrarily assigned color of their respective modules at the bottom. Right: Heatmap of all modules (*x*-axis) identified via WGCNA in (**b**) SCx and (**d**) thalamus and their corresponding Pearson correlation to phenotypic traits (*y*-axis). Each block in the heatmap shows the direction (red: positive, blue: negative), strength (top coefficient) and significance (in brackets) of the Pearson correlation of the given module to the GAERS strain, seizure parameters, and behavioral outcomes.

**Figure 5 ijms-23-06063-f005:**
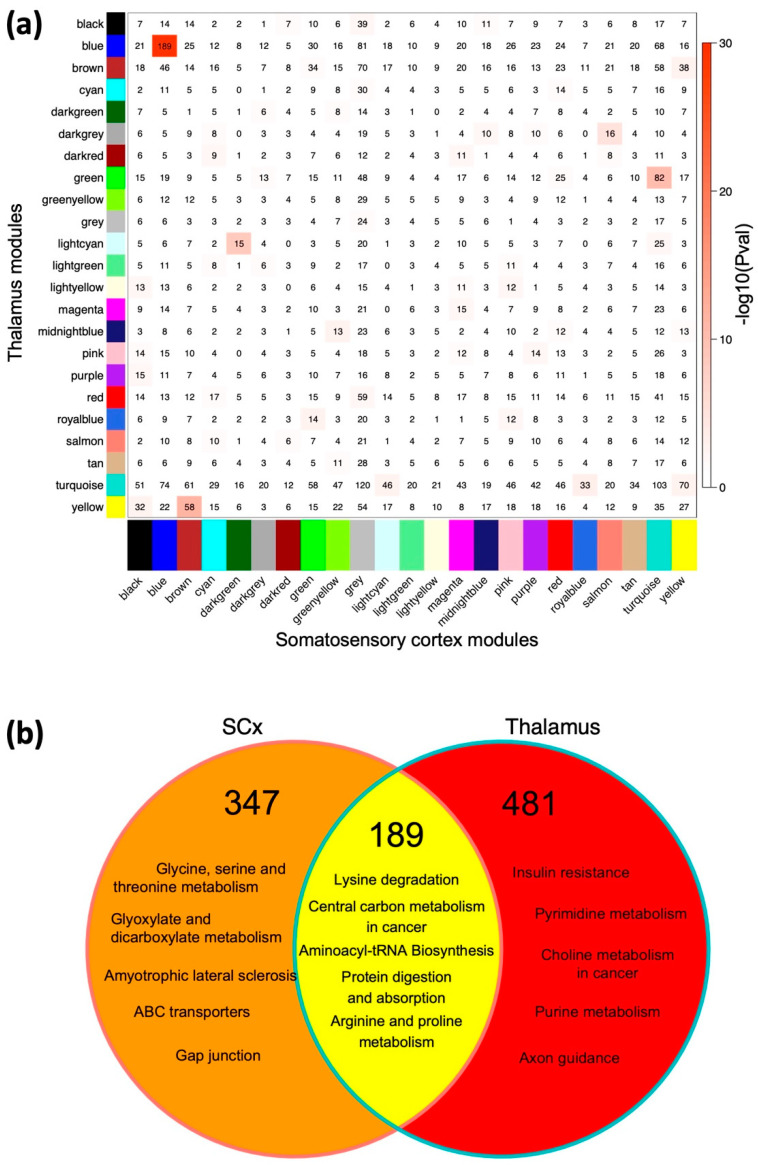
The overlap between cortical and thalamic modules. (**a**) A contingency table of overlapping proteins/metabolites between all pairs of cortical (*x*-axis) and thalamic (*y*-axis) modules. Each block in the table shows the number of overlapping proteins + metabolites in the intersection of corresponding cortical and thalamic modules. The table is color-coded with −log10 of the *p* value associated with the Fisher exact test. (**b**) A Venn diagram depicting the enriched pathways as determined by the joint pathway enrichment analysis of the region-specific and common/overlapping proteins + metabolites from the seizure-associated Blue modules. The diagram is color-coded with the number of region-specific and common proteins + metabolites in the two Blue modules.

**Figure 6 ijms-23-06063-f006:**
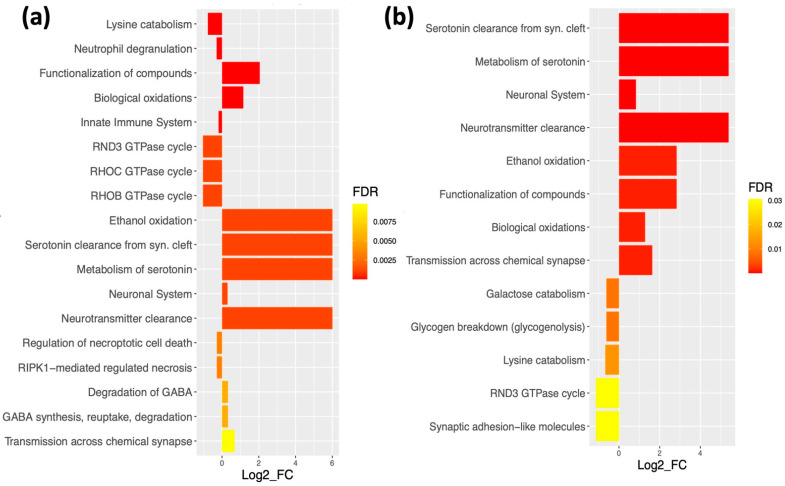
**Enrichment analysis of proteins from seizure-associated Blue modules.** Differentially regulated pathways in the Blue module from (**a**) the SCx and (**b**) thalamus identified via PADOG analysis. Significance threshold was set to FDR < 0.05. The log_2_FC (*x*-axis) indicates the direction of regulation (log_2_FC > 0 upregulated, log_2_FC < 0 downregulated) of the whole pathway in GAERS relative to NEC.

**Table 1 ijms-23-06063-t001:** The top 10 differentially expressed proteins identified in the somatosensory cortex of the GAERS compared to the NEC rats.

Accession ID	Protein	Description	Fold Change	FDR
Q6AZ33	BLVRA	Biliverdin reductase A	−6.13954	1.02 × 10^−10^
Q6P2A7	FLOT	Flotillin	−2.05081	1.60 × 10^−9^
G3V6T7	PDIA4	Protein disulfide isomerase family A, member 4	5.357599	2.55 × 10^−9^
CAMKV	CAMKV	CaM kinase-like vesicle-associated	−6.52286	5.88 × 10^−9^
A0A0G2JXT3	FDPS	Farnesyl diphosphate synthase	−3.89937	5.88 × 10^−9^
G3V983	GSTM1	Glutathione S-transferase mu 1	4.594248	1.78 × 10^−8^
NIT2	NIT2	Nitrilase family, member 2	−1.41015	1.78 × 10^−8^
ALDH2	ALDH2	Aldehyde dehydrogenase 2 family member	6.007907	3.39 × 10^−8^
KAD1	AK1	Adenylate kinase isoenzyme 1	1.421165	2.03 × 10^−7^
A0A0G2JSW3	HBB	Haemoglobin subunit beta	7.73431	3.46 × 10^−7^

**Table 2 ijms-23-06063-t002:** The top 10 differentially expressed proteins identified in the thalamus of the GAERS compared to the NEC rats.

Accession ID	Protein	Description	Fold Change	FDR
ALDH2	ALDH2	Aldehyde dehydrogenase 2 family member	5.39302723	7.34 × 10^−13^
G3V6T7	PDIA4	Protein disulfide isomerase family A, member 4	4.98460727	5.74 × 10^−12^
Q63011	NA (fragment)	Zero beta-globin	−3.4885391	3.25 × 10^−10^
CAMKV	CAMKV	CaM kinase-like vesicle-associated	−5.5727073	3.67 × 10^−9^
A0A0G2JSW3	HBB	Haemoglobin subunit beta	8.40519465	3.75 × 10^−9^
NIT2	NIT2	Nitrilase family, member 2	−1.4379458	6.72 × 10^−9^
A0A0G2JXT3	FDPS	Farnesyl diphosphate synthase	−4.4110211	2.54 × 10^−8^
KAD1	AK1	Adenylate kinase isoenzyme 1	1.49689657	4.63 × 10^−8^
M0R544	GAA	Glucosidase, alpha, acid	3.66093389	4.65 × 10^−8^
Q6AZ33	BLVRA	Biliverdin reductase A	−6.0399292	4.65 × 10^−8^

**Table 3 ijms-23-06063-t003:** The significantly enriched pathways identified in the Blue seizure-associated module in somatosensory cortex and thalamus.

**Somatosensory Cortex**		
**Pathway**	***p*-Value**	**FDR**
Aminoacyl-tRNA biosynthesis	1.38 × 10^−6^	0.000451
ABC transporters	7.02 × 10^−6^	0.001145
Protein digestion and absorption	4.85 × 10^−5^	0.005269
Lysine degradation	7.88 × 10^−5^	0.006423
Glycine, serine and threonine metabolism	0.000158	0.009242
Arginine and proline metabolism	0.00017	0.009242
Amyotrophic lateral sclerosis (ALS)	0.000314	0.013445
Central carbon metabolism in cancer	0.00033	0.013445
**Thalamus**		
**Pathway**	***p*-Value**	**FDR**
Lysine degradation	8.29 × 10^−6^	0.002703
ABC transporters	7.20 × 10^−5^	0.009727
Aminoacyl-tRNA biosynthesis	8.95 × 10^−5^	0.009727

## Data Availability

Proteomics data is available via ProteomeXchange with dataset identifier PXD033987. Metabolomics data is openly available at https://store.erc.monash.edu/experiment/view/15139/, accessed on 1 March 2022.

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
