# Peer review of "An Integrated Multi-Omic Network Analysis Identifies Seizure-Associated Dysregulated Pathways in the GAERS Model of Absence Epilepsy"

_ijms, 2022, doi:10.3390/ijms23116063_

Round 1

Reviewer 1 Report

The present article has serious flaws. For example, the abstract lacks the methods, results, and discussion. There are no Table 1 and Table 2 in the main text. Inappropriate supplemental information. There is the wrong title for the supplementary table. There is no data in supplementary table 5, supplementary table 6.

Results

Figure 1a. no EEG trace of the NEC rats.

Figure 1. the authors did not descript the statistical methods, the authors should perform multiple tests for correcton.

Lack the validation data. For example, the authors should verify the differentially expressed proteins using the immunoblotting assay.

An inconsistent reference format.

Author Response

Dear reviewer, 

Thank you for your valuable comments and suggestions regarding our study. We have addressed them in the revised manuscript (attached). Additionally, we would like to elaborate on each point below for your consideration:

  • The present article has serious flaws. For example, the abstract lacks the methods, results, and discussion.
  • It seems that several technical issues have occurred during the automatic manuscript reformatting and IJMS-approved template generation. The abstract and references did not carry over well, resulting in loss of text in several places. We have manually corrected the formatting and the missing text.
  • There are no Table 1 and Table 2 in the main text.
  • Same formatting issue as above. Table 1 and Table 2 have been embedded into the main text.
  • Inappropriate supplemental information. There is the wrong title for the supplementary table. There is no data in supplementary table 5, supplementary table 6.
  • Same technical issues as above. A compiled zip file has been uploaded which included supplementary tables 1-6.

Results

  • Figure 1. the authors did not descript the statistical methods, the authors should perform multiple tests for correcton.
  • Additional details elaborating on the basic statistical tests used for the behavioral data analysis have been added to the methods section 4.5 (lines 475-482).
    We are concerned that the reviewer may have felt that the statistical methods had not been well described. We do, however, feel that the manuscript does provide sufficient information on each of the methods applied. Correction for multiple tests is commonly used when a large number of independent tests have been applied, and significant results might have occurred by chance. In this manuscript, a single t-test per phenotypic outcome was applied and thus correction for multiple tests is not required.
    Regarding the high-thoughput data analysis, we would like to draw the reviewer's attention to lines 598-600:

For both proteomics and metabolomics, p-values associated with the t-tests were corrected for multiple comparisons using the Benjamini-Hochberg method, and significance threshold was set to FDR<0.05”

Similarly, with the WAGNA method and consensus modules and relating modules to sample traits we alleviate the need for adjustment for multiple testing (see https://bmcbioinformatics.biomedcentral.com/articles/10.1186/1471-2105-9-559).

  • Lack the validation data. For example, the authors should verify the differentially expressed proteins using the immunoblotting assay.
  • We would like to acknowledge that immunoblotting assays have some utility in identifying differentially expressed proteins. However, it should be recognised that LC-MS/MS has several important advantages over traditional assays for analytical sequencing. Firstly, LC-MS/MS’s sensitivity in detecting lower weight molecular compounds is significantly superior compared to immunoblotting assays. Secondly, immunoblotting assays often suffer from false positive or negative results due to antibody performance. These false results could be avoided using next-generation high-throughput mass spectrometry-based methods such as LC-MS/MS which provide increased detection accuracy. Among other such articles, LC-MS/MS’s advantage over conventional protein detection methods are further detailed in: 
      • Grebe SK, Singh RJ. 2011 https://pubmed.ncbi.nlm.nih.gov/21451775/
      • Hoofnagle, A.N.; Wener, M.H, 2009 https://pubmed.ncbi.nlm.nih.gov/19538965/

Additionally, we would also like to acknowledge that while validation can also be performed via immunoblotting assays, these assays once again suffer from several drawbacks. As suggested by Neilson et al: (https://pubmed.ncbi.nlm.nih.gov/21243637/ ) immunoblots can be expensive and low throughput. Furthermore, it is unrealistic, and thus not common to conduct immunoblot assays on all differentially expressed proteins. Due to the substantial number of differentially expressed proteins identified, conducting immunoblotting assays would require considerable amount of brain tissue as well as time and effort. Since the main purpose of our study was to identify differences on how proteins interact in groups/modules (as opposed to individually), validation of any specific proteins is less meaningful in this context. 

We once again thank you for your time and hope that we have addressed all of the raised points. Please find attached the revised manuscript file with tracked changes as well as an excel file with supplementary tables 1-6. 

Best regards,

A. Harutyunyan, on behalf of all authors. 

Reviewer 2 Report

The manuscript describes an attempt to investigate proteome and metabolome signatures associated with absence epilepsy in an animal model. The study is interesting and the limitations of translating their study in human samples are reported. However, in the Discussion authors should discuss the importance of integrating their data with genomic and transcriptome information, since a real "multi-omic approach" should include also these data. Proteome signatures are strictly linked to transcriptome and this connection may unveil intriguing mechanistic insights (for instance, influence of non-coding RNAs or post-translational modifications, epigenetic effects, that may represent potential therapeutical targets). 

Author Response

Dear reviewer, 

Thank you for your valuable comments and suggestions regarding our study. We have addressed them in lines 388-392 (quoted below) in the attached revised manuscript. 

"The approach we applied allowed us to evaluate modules informed by both proteomic and metabolomic data. Regrettably, we lack transcriptomic data for further omics integration analyses. It can be noted, however, that little correlation has been found between the proteome and the transcriptome due to various downstream modifications including post-translational modifications or alternative splicing".

Please find attached the revised manuscript files with tracked changes as well as a an excel file with supplementary tables 1-6. 

Thanks and regards,

A. Harutyunyan on behalf of all authors.
